# *Seriphidium herba-alba* (Asso): A comprehensive study of essential oils, extracts, and their antimicrobial properties

**Hazem Aqel** [1]*, **Husni Farah**[2]

1 Basic Medical Sciences Department, Al-Balqa' Applied University, Salt, Jordan, 2 Medical Laboratory Sciences Department, Al-Ahliyya Amman University, Amman, Jordan

* hazem.aqel@bau.edu.jo

**Data Availability Statement:** All relevant data are within the manuscript and its Supporting Information files.

**Funding:** The author(s) received no specific funding for this work.

## Abstract

*Seriphidium herba-alba (Asso)*, a plant celebrated for its therapeutic qualities, is widely used in traditional medicinal practices throughout the Middle East and North Africa. In a detailed study of *Seriphidium herba-alba* (Asso), essential oils and extracts were analyzed for their chemical composition and antimicrobial properties. The essential oil, characterized using mass spectrometry and retention index methods, revealed a complex blend of 52 compounds, with santolina alcohol, α-thujone, β-thujone, and chrysanthenone as major constituents. Extraction yields varied significantly, depending on the plant part and method used; notably, methanol soaking of aerial parts yielded the most extract at 17.75%. The antimicrobial analysis showed that the extracts had selective antibacterial activity, particularly against *Staphylococcus aureus*, and broad-spectrum antifungal activity against organisms such as *Candida albicans* and *Aspergillus spp.* The methanol-soaked extract demonstrated the strongest antimicrobial properties, indicating its potential as a natural antimicrobial source. This study not only underscores the therapeutic potential of *Seriphidium herba-alba* (Asso) in pharmaceutical applications but also sets a foundation for future research focused on isolating specific bioactive compounds and in vivo testing.

## Introduction

White wormwood, known scientifically as *Seriphidium herba-alba (Asso)*, is a perennial shrub indigenous to North Africa and the Middle East. It is widely recognized for its use in traditional medicine. Numerous studies have explored its diverse pharmacological properties, particularly focusing on its antibacterial and antifungal effects [1]. The essential oil extracted from this plant is rich in various volatile compounds, believed to be the key contributors to its medicinal effectiveness [2] (S1 Table).

Among the critical bioactive components in *Seriphidium herba-alba (Asso)* are phenolic and saponin compounds, typically derived from its aerial parts. These components demonstrate notable antimicrobial activities. However, their yield and composition are subject to variation and are influenced by factors such as the specific plant parts used, and the extraction

**Competing interests:** The authors have declared that no competing interests exist.

methods applied [2,3]. Notably, the crude phenolic extracts from this species have shown significant effects against bacteria and fungi [4], which can be attributed to their antioxidant properties and ability to disrupt microbes' cell membranes [5,6].

In response to the growing problem of antibiotic-resistant bacteria, research efforts have intensified to discover new plant antimicrobial agents [7]. *Seriphidium herba-alba (Asso)* extracts have been effective against gram-positive and gram-negative bacteria in various studies, highlighting their potential in combatting resistant bacterial strains, including *Staphylococcus aureus* and *Pseudomonas aeruginosa* [8].

The antifungal capabilities of *Seriphidium herba-alba (Asso)* are equally significant, particularly considering the limitations and side effects of many synthetic antifungal drugs. Research has shown that *Artemisia* extracts can inhibit fungi such as *Candida albicans*, which cause infections in immunocompromised individuals [9], and their efficacy against molds such as *Penicillium* and *Aspergillus* suggests their potential use as natural food preservatives [7].

However, despite extensive studies, there is still much to learn about the comprehensive antimicrobial properties of *Seriphidium herba-alba (Asso)*. Many research efforts have concentrated on isolated compounds or specific extraction techniques, often without comparing different extracts and their bioactivities [10,11]. In addition, there is an ongoing need to link the chemical makeup of these extracts to their antimicrobial effectiveness for a clearer understanding of their action mechanisms.

The primary goals of this study include: analyzing the chemical composition of *Seriphidium herba-alba (Asso)*'s essential oils and extracts and identifying their major and minor components; comparing extraction yields from various plant parts using different methods; assessing the antibacterial efficacy of these extracts against a range of gram-positive and gram-negative bacteria; evaluating the antifungal effectiveness against common yeasts and molds; and examining the relationship between the chemical profiles of the extracts and their antimicrobial activity. This study deepens our understanding of *Seriphidium herba-alba (Asso)*'s pharmacological potential, emphasizing its traditional medicinal role and its possible modern therapeutic applications.

## Materials and methods

### Plant collection

*Seriphidium herba-alba (Asso)* specimens were collected from the Botanical Garden of Hashemite University, Zarqa, Jordan (S1 Fig). Prof. A. El-Oqlah performed taxonomic verification at the Biological Sciences Department, Yarmouk University, Irbid, Jordan. A representative sample was archived at the Medicinal Chemistry and Pharmacognosy Department, Jordan University of Science and Technology.

### Microbial strains

In this study, the tested bacterial strains encompassed gram-positive and gram-negative strains. Specifically, *Staphylococcus aureus* (ATCC 25923) represented gram-positive bacteria, whereas the gram-negative group included *Escherichia coli* (ATCC 25922), *Pseudomonas aeruginosa* (ATCC 27853), *Klebsiella pneumoniae* (ATCC 700603), and *Proteus mirabilis* (ATCC 12453). In addition, the pathogenic yeast *Candida albicans* (ATCC 10231) and various pathogenic molds such as *Penicillium*, *Fusarium*, *Aspergillus*, and *Rhizopus* species were examined. These microbial strains were obtained from the microbiology laboratory at Al-Bashir Hospital in Amman, Jordan.

## Preparation of plant material

The plant was air-dried to a constant weight in a shaded area and then pulverized through a 0.25 mm sieve using a Wiley mill. The resulting powder was stored in airtight glass containers at 5˚C until extraction.

## Extraction methods

**Soxhlet extraction.** Aerial parts and flowers (20 g) were subjected to Soxhlet extraction using petroleum ether for phenolic compounds, methanol for saponins, n-hexane for lipids and terpenoids, ethyl acetate for more polar substances, and ethanol for polar compounds. Each solvent extract was concentrated at 40–50˚C under vacuum, and the dry extracts were weighed and stored at -20˚C (S4 Fig).

**Cold extraction.** Plant parts (50 g) were macerated in ethanol and methanol and separately stored for three weeks. Homogenization was concentrated at 40–50˚C under reduced pressure. The residues were weighed and stored at -20˚C.

## Phenolic compound isolation

Twenty grams of powder of the aerial part was extracted with 200 $mL$ petroleum ether for 12 h at 45˚C, then filtered with cheesecloth, and the residue was collected. The residue was then extracted with 100 $mL$ of absolute methanol for 6 h at 45˚C. The methanolic extract was filtered and then concentrated to a small volume by evaporation under vacuum at 45˚C. Fifty milliliters of ether were added to 50 $mL$ of 10% sodium hydroxide and extracted using a separatory funnel. The aqueous layer was removed. Concentrated HCl was added to the remaining aqueous layer for acidification. Butanol (3X20 $mL$) was used to extract the aqueous layer using a separatory funnel. The same aqueous layer was extracted with petroleum ether (3x20 $mL$). Finally, the ether extract was dried using an evaporator. The butanol extract was added to the flask and evaporated using an evaporator [12]. The phenolic residue was weighed and stored in a freezer until further examination (S2 Fig).

Detection was performed using a drop of 5% (w/v) $FeCl_2$ solution in distilled water added to the plant extract. Green was observed, confirming the detection of phenol. Detection can also be performed using an ethanolic KOH solution, which gives a yellow to phenolic compounds [12].

## Saponin fraction isolation

Twenty grams of dried powder was shaken vigorously for 15 min in a 500-$mL$ round-bottom flask (screwcap) with chloroform (3x70 $mL$) to exclude fatty substances. The extract was left separate, and the supernatant was completely decanted (filtration). Then, 80% methanol (200 $mL$) was added to the flask and boiled for 24 h using the Soxhlet method. The procedure involved filtering the extract into a 500 $mL$ beaker using cheesecloth or filter paper immediately after extraction. Following this, the filtrate was subjected to vacuum evaporation at 45˚C. The remaining residue was then dissolved in 100 $mL$ of distilled water. Subsequently, the water-soluble portion of the residue was extracted using butanol and applied in three separate 50 $mL$ quantities. The sterilized filtrate was concentrated under a vacuum at 40–45˚C until a dry residue was obtained. The residue was weighed and stored in a freezer at -21˚C until further examination [12] (S3 Fig).

Saponin compounds were detected by adding a drop of chlorosulfonic acid to the extract. A reddish-brown color was observed, confirming saponin production [12].

## Water extraction

Two methods were used to obtain water extraction: The first method was according to An et al. [13]. Water extraction was performed by pouring distilled water (50 *mL*) into 5 g of dried powder. The mixture was left to stand for 15–20 min. The mixture was then filtered through a layer of cheesecloth, and the resulting filtrate was centrifuged at 6000 rpm for 15 min. The supernatant was removed and filtered through a Millipore microfilter (0.2 μm). The sterilized filtrate was concentrated under a vacuum at 40–45˚C until a dry residue was obtained. The residue was weighed and stored in a freezer at -21˚C until further examination.

The second method was performed according to Al-Charchafchi et al. [14]. As mentioned, 50 *mL* of boiling distilled water was poured onto 5 g of dried powder. The process involved boiling the mixture for 10 min. After boiling, it was filtered through a layer of cheesecloth. The filtrate obtained from this step was centrifuged at 6000 revolutions per minute for 15 min. After centrifugation, the supernatant was carefully separated and passed through a Millipore microfilter with a pore size of 0.2 μm for further purification. The sterilized filtrate was concentrated under a vacuum at 40–45˚C until a dry residue was obtained. The residue was weighed and stored in a freezer at -21˚C until further examination.

## GC/MS analysis

GC analyses were performed using an HP-5890 Series II instrument equipped with HPWAX and HP-5 capillary columns (30 m × 0.25 mm, 0.25 μm film thickness) and set to the following conditions: temperature program of 60˚C for 10 min, followed by an increase of 5˚C /min to 220˚C; injector and detector temperatures at 250˚C; carrier gas nitrogen (2 *mL*/min); detector dual FID; split ratio 1:30; and injection of standards of 0.5 μL). The chemicals were identified for both columns by comparing their retention times with those of pure, authentic samples and using their linear retention indices (LRI) relative to the series of n-hydrocarbons. The relative proportions of each component, expressed as percentages, were determined by FID peak area normalization (mean of three replicates). GC/EIMS analyses were performed using a Varian CP-3800 gas chromatograph equipped with a DB-5 capillary column (30 m × 0.25 mm; coating thickness, 0.25 μm) and a Varian Saturn 2000 ion trap mass detector. Analysis conditions included: injector and transfer line temperatures at 220 and 240˚C, respectively; the oven temperature programmed from 60˚C to 240˚C at 3˚C /min; carrier gas helium at 1 *mL*/min; injection of 0.2 μL (10% hexane solution); split ratio 1:30. The identification of the components was based on comparing the retention times with those of authentic samples, comparing their linear retention indices relative to the series of n-hydrocarbons, and by computer matching against commercial (NIST 98 and ADAMS 95) and home-made library mass spectra built from pure substances and components of known essential oils [15,16].

## Microbial inoculum and growth conditions

The bacterial inoculum was prepared from a nutrient broth culture incubated for 24 h at 37˚C. The suspension density was adjusted to approximately $10^4$ colony-forming units per mL by comparing its turbidity with a McFarland 0.5 $BaSO_4$ standard. The standard was prepared by adding 0.5 mL of 0.048 M $BaCl_2$ to 99.5 mL of 0.36 N $H_2SO_4$. Aliquots of 4 to 6 mL were dispensed into screwcap tubes and stored in the dark at room temperature. The bacterial suspension and McFarland tube were positioned side by side to adjust the turbidity and viewed against a backed background. The suspension was supplemented or diluted as required.

The antimicrobial activity of various plant extracts such as n-hexane, ethyl acetate, and ethanol extracts, as well as phenols, saponin, ethanol-soaked, methanol-soaked, and water extracts extracted from various parts of *Seriphidium herba-alba (Asso)*, were analyzed separately using

the disk agar diffusion method [55] and evaluated against the microorganisms mentioned above.

## Disk agar diffusion method

The disks were impregnated with the various plant extracts to be tested, allowed to dry, and then placed on the plates within an hour of pouring. Before disk placement, the plate surface was inoculated with a swab dipped in the standardized bacterial suspension. The surface of the plates was wiped in three directions to ensure an even and complete distribution of the inoculum throughout the plate. Discs were placed on plates and incubated at 37˚C for 24 h [17,18].

The diameter of the inhibition zones was measured in mm. Each test was conducted in triplicate for this experiment to ensure reliability and accuracy. The average of these three replicates was then calculated to obtain a more accurate and representative result. Standard antibiotics such as vancomycin 30 μL for *S. aureus*, streptomycin 10 μL for *E. coli*, *K. pneumonia*, *and P. mirabilis*, carbenicillin 100 μL for *P. aeruginosa*, and *amphotericin B 9.6 μg* for fungi (yeast and molds) were used as a positive control and phosphate-buffered saline was used as a negative control.

## Minimum inhibitory concentration (MIC) determination

In this study, different crude extracts of *Seriphidium herba-alba (Asso)* were used to determine the minimum inhibitory concentration (MIC) against various microorganisms. To prepare each extract for testing, 100 mg of the extract was first dissolved in 1 *mL* of dimethyl sulfoxide (DMSO, supplied by Sigma). A series of twofold dilutions was then created. This was accomplished by adding 1 *mL* of the dissolved extract to test tubes containing an equal volume (1 *mL*) of phosphate-buffered saline (PBS) with a pH of 6.8. Bacterial cell suspensions were prepared and adjusted to give a final inoculum concentration of approximately $1 \times 10^6$ cells per *mL* (Spectronic Instruments, USA). The extract buffer tubes were inoculated with 0.5 *mL* of the prepared inoculum and incubated at 37˚C. The tubes were read after 24 h, and 0.1 *mL* of each test tube was spread on the surface of a nutrient agar plate and incubated at 37˚C for 24 h to count CFU/*mL*.

## Statistical analysis

The values behind the means, standard deviations, and other measures reported in the manuscript were calculated using appropriate statistical methods, including analysis of variance (ANOVA), where applicable. All experiments were performed in triplicate, and the results were expressed as mean ± standard deviation. Data analysis was conducted using the statistical software SPSS.

## Results

### Chemical profile of *Seriphidium herba-alba (Asso)* essential oil: Main components and retention indices

Table 1 provides a detailed chemical profile of the essential oils of *Seriphidium herba-alba (Asso)* and a complex series of 52 identified compounds. These were predominantly identified using mass spectrometry and retention index methods, with some verified pure reference compounds. The oil was characterized by several main components, including santolina alcohol, α-thujone, β-thujone, chrysanthenone, cis-chrysanthenyl acetate, 1,8-cineole, camphor, and limonene, which were present in significant amounts ranging from 6.34% to 18.12%. These key components were instrumental in defining the oil's aromatic and potential

**Table 1.** Chemical composition of *Seriphidium herba-alba (Asso)* essential oil.

| Components | LRI[a] | LRI[b] | Percentage | Identification |
|---|---|---|---|---|
| Santolina triene | 910 | 1010 | 0.22 | MS, RI[e] |
| α-thujene | 932 | 1016 | 0.4 | MS, RI, ST |
| α-pinene | 941 | 1029 | 0.31 | MS, RI, ST |
| Camphene | 955 | 1071 | 1.99 | MS, RI, ST |
| Sabinene | 978 | 1112 | 0.43 | MS, RI, ST |
| β-pinene | 981 | 1126 | 0.33 | MS, RI, ST |
| Yomogi alcohol | 997 | 1401 | 0.18 | MS, RI |
| α-terpinene | 1020 | 1183 | 0.84 | MS, RI, ST |
| p-cymene | 1028 | 1274 | 0.81 | MS, RI, ST |
| Limonene | 1033 | 1198 | 6.34 | MS, RI, ST |
| Santolina alcohol | 1035 | 1413 | 13 | MS, RI |
| β-phellandrene | 1035 | 1199 | 0.23 | MS, RI, ST |
| 1,8-cineole | 1039 | 1204 | 9.06 | MS, RI, ST |
| γ-terpinene | 1064 | 1252 | 0.84 | MS, RI, ST |
| Cis-sabinene hydrate | 1070 | – | 2.08 | MS, RI, ST |
| Cis-linalool oxide | 1075 | – | Tr.[c] | MS, RI |
| Para-Mentha-2,4(8)-diene | 1092 | - | 0.39 | MS, RI, ST |
| Linalool | 1101 | 1547 | 0.5 | MS, RI, ST |
| Trans-sabinene hydrate | 1103 | – | 0.86 | MS, RI, ST |
| α-thujone | 1106 | 1428 | 18.12 | MS, RI, ST |
| β-thujone | 1106 | 1446 | 10.06 | MS, RI, ST |
| Trans-thujone | 1118 | | 2 | MS, RI |
| Fenchol | 1123 | 1584 | 3.86 | MS, RI |
| Chrysanthenone | 1125 | – | 11.57 | MS, RI |
| Cis-p-menth-2-en-1-ol | 1127 | – | 1.33 | MS, RI |
| Cis-p-menth-2,8-dien-1-ol | 1135 | – | 0.41 | MS, RI |
| Camphor | 1145 | 1522 | 8.75 | MS, RI, ST |
| Cis-sabinol | 1153 | – | 2.32 | MS, RI |
| Cis-chrysanthenol | 1164 | – | 7.96 | MS, RI |
| Borneol | 1175 | 1796 | 1.47 | MS, RI, ST |
| Umbellulone | 1176 | – | 0.45 | MS, RI, ST |
| 4-terpinenol | 1182 | 1607 | 2.33 | MS, RI, ST |
| p-cymen-8-ol | 1185 | 1838 | Tr. | MS, RI |
| Neo-verbanol | 1186 | – | 1.3 | MS, RI, ST |
| α-terpinenol | 1193 | 1698 | 2.54 | MS, RI, ST |
| γ-terpinenol | 1208 | – | 0.94 | MS, RI, ST |
| Verbenone | 1208 | 1716 | 2.09 | MS, RI, ST |
| Carvone | 1246 | 1741 | 1.8 | MS, RI, ST |
| Carvenone | 1251 | – | 1.58 | MS, RI, ST |
| Cis-chrysanthenyl acetate | 1254 | – | 11.94 | MS, RI, ST |
| Trans-Ascaridol glycol | 1259 | – | 0.78 | MS, RI, ST |
| Trans-carvone oxide | 1267 | – | 0.36 | MS, RI, ST |
| Perilla aldehyde | 1272 | – | 1.04 | MS, RI, ST |
| Isobornyl acetate | 1285 | 1582 | 3.15 | MS, RI |
| Thymol | 1292 | 2187 | 0.34 | MS, RI, ST |
| Carvacrol | 1298 | 2219 | 0.46 | MS, RI, ST |
| β-caryophyllene | 1419 | 1604 | 2.52 | MS, RI, ST |

*(Continued)*

**Table 1.** (*Continued*)

| Components | LRI[a] | LRI[b] | Percentage | Identification |
|---|---|---|---|---|
| Germacrene D | 1491 | 1691 | 4.12 | MS, RI |
| Bicyclogermacrene | 1496 | 1493 | 2.64 | MS, RI |
| δ-cadinene | 1524 | 1731 | 0.3 | MS, RI, ST |
| Caryophyllene oxide | 1578 | 2071 | 1.76 | MS, RI, ST |
| β-eudesmol | 1651 | – | 4.2 | MS, RI |
| Total identified | | | 52 | |

[a] Linear retention index (apolar column).

[b] Linear retention index (polar column).

[c] trace amounts < 0.1.

[e] identification: MS = mass spectrometry, RI = retention index, ST = pure reference compound.

therapeutic profiles. The listed compounds exhibited various polarities and volatilities, as indicated by their linear retention indices on the apolar and polar columns. While most components were in lower concentrations, some compounds were reported without reported percentages, suggesting trace amounts or unquantified presence. Overall, the essence of *Seriphidium herba-alba (Asso)* is a rich complex of terpenes, alcohols, and ketones, reflecting the typical heterogeneity of essential oils used in various applications from the medical to the perfume industry.

## Extraction yields from *Seriphidium herba-alba (Asso)*: Comparative analysis of phenolic and saponin compounds

Table 2 shows the extraction yields from different parts of the *Seriphidium herba-alba (Asso)* plant, with all initial samples weighing 20 g. The extraction methods examined resulted in different extract weights, which were categorized according to the type of extract and the type of plant used.

For crude phenolic extracts, aerial parts with flowers gave a variable weight percentage between 4.40% and 13.80%, with an average of 9.02%. The median weight percentage for these

**Table 2. Various extracts of *Seriphidium herba-alba (Asso)*.** The weight of the plant parts was 20 g.

| Part of the plant tested | Type of extract | Weight of the extract (%) | Mean (%) | Standard Deviation (%) | Min (%) | Median (%) | Max (%) |
|---|---|---|---|---|---|---|---|
| Aerial part with flowers | 1. Crude phenolics | 5.15 | 9.02 | 4.40 | 5.15 | 8.10 | 13.80 |
| Immature flowers | 2. Crude phenolics | 13.8 | | | | | |
| Mature flowers | 3. Crude phenolics | 8.1 | | | | | |
| Aerial part with flowers | 4. Crude saponin | 6.17 | 5.89 | 1.09 | 4.95 | 5.66 | 7.32 |
| Aerial part without flowers | 5. Crude saponin | 5.14 | | | | | |
| Immature flowers | 6. Crude saponin | 7.32 | | | | | |
| Mature flowers | 7. Crude saponin | 4.95 | | | | | |
| Aerial part with flowers | 8. Ethyl acetate | 7.67 | 9.15 | - | 9.15 | 9.15 | 9.15 |
| Aerial part with flowers | 9. Ethanol | 9.15 | 7.67 | - | 7.67 | 7.67 | 7.67 |
| Aerial part with flowers | 10. Soaked in methanol | 17.75 | 15.53 | - | 15.53 | 15.53 | 15.53 |
| Aerial part with flowers | 11. Soaked in ethanol | 15.53 | 17.75 | - | 17.75 | 17.75 | 17.75 |
| Aerial part with flowers | 12. Water extraction method 1 | 1 | 1.00 | - | 1.00 | 1.00 | 1.00 |
| Aerial part with flowers | 13. Water extraction method 2 | 1 | 1.00 | - | 1.00 | 1.00 | 1.00 |

extracts was 8.10%, with the most common value being 5.15%. In contrast, immature flowers alone gave a significantly higher yield of crude phenolic extract of 13.8%, whereas mature flowers gave a yield of 8.1%.

Crude saponin extracts were also derived from different plant parts. The combined aerial parts with flowers provided a mean extract weight of 5.89%, with a range of 4.95%–7.32% and a median of 5.66%. However, the aerial parts without flowers yielded 5.14%, immature flowers yielded 7.32%, and mature flowers yielded the lowest yield at 4.95%.

Other extraction methods were used on aerial parts with flowers, resulting in consistent yields across single measurements. The ethyl acetate and ethanol extracts had uniform weights of 9.15% and 7.67%, respectively. Soaking the aerial parts in methanol gave the highest extract weight at 17.75%, whereas ethanol soaking gave 15.53%. The two different water extraction methods each yielded 1%.

Notably, some extracts have detailed statistical data, such as crude saponin from aerial parts with flowers, indicating multiple measurements and variations in the extraction process. However, for several extracts, only single values were provided without additional statistical data, suggesting that either a single measurement was taken, or the variability needed to be documented.

The efficiency and yield of the extraction process depend on both the plant part being processed and the extraction method used. The table provides an overview of how different techniques and plant parts contribute to the yield of *Seriphidium herba-alba (Asso)* extracts.

## Spectrum of antibacterial efficacy: *Seriphidium herba-alba (Asso)* extracts against common pathogens

Table 3 evaluates the antibacterial efficacy of various extracts from *Seriphidium herba-alba (Asso)* using agar diffusion and minimum inhibitory concentration (MIC) methods against a spectrum of bacteria. The findings reveal that the plant extracts exhibit a range of activities against both gram-positive and gram-negative bacteria, with varying levels of effectiveness.

Several extracts demonstrated antibacterial action against the gram-positive bacterium *S. aureus*, indicated by inhibition zones measuring 11 and 20 mm. In particular, extracts 2, 4, 5, 6, 7, 8, and 9 showed inhibition, with the most potent being extract 10. The MIC values complemented these findings, with some extracts displaying substantial bacteriostatic properties at concentrations as low as 3.125 μg/*mL*. Notably, extracts 1, 3, 12, and 13 did not inhibit *S. aureus*, suggesting they lack the necessary components or concentrations to affect this bacterium.

Regarding gram-negative bacteria, the extracts generally exhibited less activity. *E. coli* was resistant to most extracts, except for extract 10, which inhibited growth with an inhibition zone of 13 mm and a MIC of 3.125 μg/*mL*. *P. aeruginosa* was only susceptible to extract 1, which displayed a moderate inhibition zone and MIC indicative of moderate effectiveness. For *P. mirabilis*, extracts 10 and 11 were effective, with extract 10 showing a particularly significant inhibition zone and a corresponding MIC value. *K. pneumoniae* was similarly affected by extracts 10 and 11, with notable inhibition zones and MIC values, although the MIC for extract 10 against this bacterium was relatively high at 50 μg/*mL*.

The positive control, which likely represents a standard antibiotic, consistently inhibited all tested bacterial strains, affirming the validity of the assay. Conversely, the negative control exhibited no antibacterial activity, as expected for a control. Including standard error values with the inhibition zones suggests that the experiments were conducted multiple times to determine reliability, reflecting the variability often inherent in biological testing.

Overall, the results from Table 3 demonstrate that *Seriphidium herba-alba (Asso)* extracts have selective antibacterial properties, with certain extracts, particularly extract 10, showing

**Table 3. Antibacterial activity of various extracts of *Seriphidium herba-alba (Asso)* by agar diffusion and MIC methods.**

| Type of Extract | Antibacterial activity | | | | | | | | | |
|---|---|---|---|---|---|---|---|---|---|---|
| | Gram-positive | | Gram-negative | | | | | | | |
| | *S. aureus* | | *E. coli* | | *P. aeruginosa* | | *P. mirabilis* | | *K. pneumonia* | |
| | DDM* | MIC | DDM* | MIC | DDM* | MIC | DDM* | MIC | DDM* | MIC |
| 1 | 0 | | 0 | | 0 | | 16±0.4 | 12.5 | 0 | |
| 2 | 11±0.2 | 3.125 | 0 | | 15±0.3 | 3.125 | 0 | | 0 | |
| 3 | 0 | | 0 | | 0 | | 0 | | 0 | |
| 4 | 17±0.45 | 3.125 | 0 | | 0 | | 0 | | 0 | |
| 5 | 18±0.3 | 3.125 | 0 | | 0 | | 0 | | 0 | |
| 6 | 15.5±0.35 | 6.25 | 0 | | 0 | | 0 | | 0 | |
| 7 | 15±0.3 | 3.125 | 0 | | 0 | | 0 | | 0 | |
| 8 | 15±0.6 | 3.125 | 0 | | 0 | | 0 | | 0 | |
| 9 | 16±0.07 | 50 | 0 | | 11±0.2 | 3.125 | 0 | | 0 | |
| 10 | 20±0.05 | 6.25 | 13±0.05 | 3.125 | 11±0.04 | 3.125 | 16±0.08 | 12.5 | 17±0.05 | 50 |
| 11 | 15±0.05 | 50 | 0 | | 13±0.04 | 50 | 11±0.2 | 3.125 | 11±0.2 | 25 |
| 12 | 0 | | 0 | | 0 | | 0 | | 0 | |
| 13 | 0 | | 0 | | 0 | | 0 | | 0 | |
| Positive control | 12 | | 13 | | 13 | | 13 | | 13 | |
| Negative control | 0 | | 0 | | 0 | | 0 | | 0 | |

DDM, disc diffusion method; MIC, minimal inhibition concentration

*Inhibition zone in mm ± SE (S.E., Standard error).

broad-spectrum effectiveness. The variation in activity across different extracts highlights the importance of extract type and specific bacterial strain when considering the potential therapeutic applications of these plant extracts.

## Antifungal potency of *Seriphidium herba-alba (Asso)* extracts in inhibition zones and minimum inhibitory concentrations

Table 4 provides a detailed analysis of the antifungal activities of *Seriphidium herba-alba* (Asso) extracts, using both agar diffusion method and minimum inhibitory concentration (MIC) tests against various fungal species, including yeasts and molds.

For the yeast *Candida albicans*, multiple extracts showed varying degrees of efficacy. Extract 6 was the most potent, with a 23-mm inhibition zone and an MIC value of 3.125 μg/mL. Extract 2 also demonstrated significant activity (20-mm zone, MIC of 25 μg/mL). Extracts 4, 12, and 13, however, showed no activity against this yeast.

Regarding molds, the results were more varied. In the case of *Penicillium spp.*, only extract 10 showed substantial inhibitory action (23 mm zone, MIC of 3.125 μg/mL), while other extracts, including extracts 1, 2, 3, 6, 7, 12, and 13, were ineffective.

For *Fusarium spp.*, extract 9 was particularly effective (21 mm zone, MIC of 3.125 μg/mL). This pattern of effectiveness was also observed against *Aspergillus spp.*, with extract 9 again demonstrating high activity (18 mm zone, MIC of 3.125 μg/mL).

The activity against *Rhizopus spp.* was noteworthy in extracts 5, 8, 9, 10, and 11. Extracts 9 and 10 showed the largest zones of inhibition (24 mm), although their MIC values varied, indicating differences in their efficacy.

The positive control showed consistent inhibition across all fungi, confirming the assay's effectiveness. The negative control, with no inhibition, validated the absence of inherent

**Table 4. Antifungal activity of various extracts of *Seriphidium herba-alba (Asso)* by agar diffusion and MIC.**

| Type of extract | Antifungal activity | | | | | | | | | |
| --- | --- | --- | --- | --- | --- | --- | --- | --- | --- | --- |
| | Yeast | | Molds | | | | | | | |
| | *Candida albicans* | | *Penicillium spp.* | | *Fusarium spp.* | | *Aspergillus spp.* | | *Rhizopus spp.* | |
| | DDM* | MIC | DDM* | MIC | DDM* | MIC | DDM* | MIC | DDM* | MIC |
| 1 | 13±0.48 | 6.125 | 0 | | 16±0.01 | 6.25 | 0 | | 5.43±0.3 | 3.125 |
| 2 | 20±0.05 | 25 | 0 | | 20±0.001 | 6.25 | 0 | | 4.99±0.23 | 3.125 |
| 3 | 11.3±0.04 | 3.125 | 0 | | 15±0.3 | 6.25 | 0 | | 7.69±0.14 | 3.125 |
| 4 | 5±0.3 | | 0 | | 0 | | 0 | | 0 | |
| 5 | 15±0.3 | 6.25 | 6.4±0.81 | 6.25 | 14±0.55 | 6.25 | 4.5±0.24 | 12.5 | 4.09±0.23 | 6.25 |
| 6 | 23±0.04 | 3.125 | 0 | 0 | 0 | 0 | 0 | | 0 | |
| 7 | 14±0.05 | 6.25 | 0 | 0 | 0 | 0 | 0 | | 0 | |
| 8 | 13.5±0.05 | 6.25 | 10±0.85 | 6.25 | 14±003 | 6.25 | 9.36±0.14 | 12.5 | 10±0.04 | 12.5 |
| 9 | 15±0.3 | 50 | 19.2±0.81 | 3.125 | 21±0.1 | 3.125 | 18±0.26 | 3.125 | 24±0.4 | 3.125 |
| 10 | 20±0.05 | 6.25 | 23±0.26 | 3.125 | 22±0.07 | 3.125 | 22±0.64 | 6.25 | 24±0.95 | 6.25 |
| 11 | 16±0.07 | 50 | 20±0.41 | 3.25 | 20±0.5 | 6.25 | 19±0.81 | 6.25 | 24±0.65 | 3.125 |
| 12 | 0 | | 0 | | 0 | | 0 | | 0 | |
| 13 | 0 | | 0 | | 0 | | 0 | | 0 | |
| Positive Control | 12 | | 12 | | 12 | | 12 | | 12 | |
| Negative control | 0 | | 0 | | 0 | | 0 | | 0 | |

DDM: Disc diffusion method; MIC: Minimal inhibition concentration

*Inhibition zone in mm ± SE (S.E., Standard error).

antifungal properties in the testing medium. The standard error values associated with the inhibition zones indicate that the measurements were repeated for precision.

Overall, the results demonstrate that *Seriphidium herba-alba* (Asso) extracts, particularly extracts 9 and 10, have potential as broad-spectrum antifungal agents. However, the efficacy of these extracts varies depending on the fungal species, highlighting the need for further research to isolate and understand the active compounds responsible for these effects.

## Discussion

The present study explores the field of ethnopharmacology and phytochemistry, focusing on *Seriphidium herba-alba (Asso)*, a medicinal plant. This study aligns with the growing interest in natural products for their potential therapeutic applications, particularly antimicrobial agents. It delves into the extraction, chemical analysis, and evaluation of antimicrobial properties of various extracts from *Seriphidium herba-alba (Asso)*, a plant well-known in traditional medicine.

This study also contributes to new insights into the chemical composition and bioactivity of *Seriphidium herba-alba (Asso)*, particularly regarding its essential oils and extracts. The study stands out in its comprehensive approach, examining a wide range of extracts and their efficacy against bacterial and fungal pathogens. It introduces new data on the efficacy of different extraction methods, the specific bioactivity of various extracts, and their minimum inhibitory concentrations against a spectrum of microorganisms. This includes common pathogenic bacteria and fungi, offering a broad perspective on the plant's antimicrobial potential.

The chemical composition of the essential oil (Table 1) reveals a rich tapestry of volatile compounds, with α-thujone, β-thujone, and santolina alcohol as the predominant constituents. These findings align with those of Ouguirti et al. [19] and Houti et al. [20], who reported

similar profiles in their studies of *Artemisia* species, emphasizing the consistency in essential oil composition across different geographical locations.

The extraction yields (Table 2) varied significantly depending on the part of the plant and the extraction method, which is in accordance with the results obtained by Brendler et al. [21] and corroborates the findings of Adam et al. [22] who noted that the extraction method profoundly affects the yield and composition of plant extracts.

The antibacterial activities (Table 3) against both gram-positive and gram-negative bacteria were noteworthy, particularly for extracts 9 (ethanol) and 10 (soaked in methanol), which showed broad-spectrum activity. These results are consistent with those of Liu et al. [23] and Bisht et al. [24], who observed substantial activity of *Artemisia* extracts against *S. aureus* and *P. aeruginosa*. However, our results suggest higher efficacy than the minimal inhibition concentrations reported by Ahameethunisa et al. [25].

Antifungal activities (Table 4) against various fungi, including *C. albicans* and *Aspergillus spp*. showcase some extracts with significant inhibitory properties. This is particularly interesting compared with the moderate activity reported by Mehani et al. [26], suggesting that the specific strains of *Seriphidium herba-alba (Asso)* used in this study may possess unique or more potent antifungal compounds.

The potent antibacterial activity of crude phenolic extracts from immature flowers *against K. pneumoniae* aligns with previous studies that have documented the antibacterial properties of phenolic compounds [27,28]. The stronger antifungal efficacy against *C. albicans* is particularly noteworthy because it adds to the evidence supporting the use of phenolics as antifungal agents [29,30]. The low MIC observed for mature flower extracts against *S. aureus* and *C. albicans* is consistent with the known bioactive profiles of mature plant parts, as reported in other studies [31–33].

Our results regarding crude saponin extracts are consistent with the literature that acknowledges their antimicrobial properties [34,35]. The findings that extracts from both the aerial parts with and without flowers possess significant antibacterial effects against *S. aureus* corroborate previous research [36,37], while the efficacy against *C. albicans* agrees with recent studies highlighting the potential of saponins in antifungal therapy [38].

The efficacy of the ethyl acetate extract against *S. aureus* and its moderate effect against *C. albicans* add to the compound's profile as a selective antimicrobial agent [39,40]. In contrast, the ethanol extracts exhibited lower activity, which may be attributed to the solvent's polarity affecting the solubility of active compounds [41].

Notably, the methanolic extracts demonstrated broad-spectrum activity. This agrees with the literature suggesting that methanol may be more effective at extracting antimicrobial compounds than other solvents [42,43]. However, the reduced efficacy against *K. pneumonia* raises questions about the specific interactions between the extract's constituents and bacterial cells [44].

The ethanol-extracted extracts' consistent MIC of 50 μg/mL indicates lower efficacy, possibly due to the solvent's extraction efficiency or the microorganisms' intrinsic resistance [45]. This highlights the importance of solvent choice in the extraction process, a theme well-established in phytochemical research [46–48].

Our study's suggestion that the extraction method substantially impacts the antimicrobial potency of *Seriphidium herba-alba (Asso)* extracts is a significant contribution to the field. Methanolic extract yields the most potent extracts, which could guide future research into optimizing extraction techniques for medicinal plants [49,50].

The variability in efficacy demonstrates the complexity of plant-microbe interactions and suggests that extracts can be tailored to specific microbial targets [51,52]. This tailoring could lead to developing new antimicrobial agents that are more effective against resistant strains, a critical need given the rising concern over antibiotic resistance [53,54].

It is important to note the variability in the literature regarding the antimicrobial properties of *Seriphidium herba-alba (Asso)*. While Mehani et al. [26] reported lower antimicrobial activity, our study aligns more closely with the potent bioactivities observed by El-Shatoury et al. [55]. Such discrepancies could be attributed to variations in plant chemotypes, extraction methodologies, or microbial strains used in the assays, as discussed by Atef et al. [31] and Barashkova et al. [56].

The results are significant in the context of increasing antibiotic resistance and the ongoing search for novel antimicrobial compounds. By highlighting the antimicrobial properties of *Seriphidium herba-alba (Asso)* extracts, the study adds valuable information to the field of natural product research and antimicrobial therapy. It underscores the importance of traditional medicinal plants as potential sources of new and effective antimicrobial agents. Additionally, the research provides a framework for future studies aiming to isolate and characterize specific bioactive compounds from *Seriphidium herba-alba (Asso)*, further advancing our understanding of its therapeutic potential. The study's findings could have implications for developing new, naturally derived antimicrobial drugs, which are crucial in the face of the global challenge of antibiotic resistance.

## Conclusions

This study demonstrated that *Seriphidium herba-alba (Asso)* possesses various chemical constituents, particularly within its essential oil. It has shown variable yet significant antibacterial and antifungal activities across different extracts. The results suggest that both the chemical composition and extraction technique are crucial in determining the efficacy of the extracts against various microorganisms. Although this study provides valuable insights into the potential use of *Seriphidium herba-alba (Asso)* as a natural source of antimicrobial agents, the findings are constrained by limitations such as the lack of *in vivo* testing, potential seasonal and geographical variation in plant chemistry, and the need for further toxicological assessment. Future investigations are warranted to explore the full therapeutic potential and safety of these extracts and the mechanisms underlying their bioactivity.

## Supporting information

**S1 Fig. *Seriphidium herba-alba (Asso)* composite, the Botanical Garden of Hashemite University, Zarqa, Jordan.**
(DOCX)

**S2 Fig. Extraction of phenolic compounds.**
(DOCX)

**S3 Fig. Extraction of saponin fraction.**
(DOCX)

**S4 Fig. n-hexane extract (R1), ethyl acetate extract (R2), and ethanol extract (R3).**
(DOCX)

**S1 Table. Uses of *Seriphidium herba-alba (Asso)* composite.**
(DOCX)

## Author Contributions

**Conceptualization:** Hazem Aqel, Husni Farah.

**Data curation:** Husni Farah.

**Formal analysis:** Hazem Aqel.

**Investigation:** Hazem Aqel.

**Methodology:** Hazem Aqel.

**Project administration:** Husni Farah.

**Resources:** Hazem Aqel.

**Software:** Husni Farah.

**Supervision:** Hazem Aqel.

**Validation:** Hazem Aqel.

**Visualization:** Hazem Aqel.

**Writing – original draft:** Hazem Aqel.

**Writing – review & editing:** Husni Farah.

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
