## [Decision Letter · Decision Letter 0]

31 Jan 2024

PONE-D-23-41380Artemisia herba alba: A Comprehensive Study of Essential Oils, Extracts, and Their Antimicrobial PropertiesPLOS ONE

Dear Dr. Aqel,

Thank you for submitting your manuscript to PLOS ONE. After careful consideration, we feel that it has merit but does not fully meet PLOS ONE’s publication criteria as it currently stands. Therefore, we invite you to submit a revised version of the manuscript that addresses the points raised during the review process.

We look forward to receiving your revised manuscript.

Kind regards,

Vijay Tripathi

Academic Editor

PLOS ONE

Comments from Senior Staff Editor: We note that one or more reviewers has recommended that you cite specific previously published works. As always, we recommend that you please review and evaluate the requested works to determine whether they are relevant and should be cited. It is not a requirement to cite these works. We appreciate your attention to this request.

Journal Requirements:

2. Please amend your list of authors on the manuscript to ensure that each author is linked to an affiliation. Authors’ affiliations should reflect the institution where the work was done (if authors moved subsequently, you can also list the new affiliation stating “current affiliation:….” as necessary).

Reviewers' comments:

Reviewer's Responses to Questions

**Comments to the Author**

1. Is the manuscript technically sound, and do the data support the conclusions?

Reviewer #1: Yes

Reviewer #2: Yes

2. Has the statistical analysis been performed appropriately and rigorously? 

Reviewer #1: No

Reviewer #2: Yes

3. Have the authors made all data underlying the findings in their manuscript fully available?

Reviewer #1: Yes

Reviewer #2: Yes

4. Is the manuscript presented in an intelligible fashion and written in standard English?

Reviewer #1: No

Reviewer #2: Yes

5. Review Comments to the Author

Reviewer #1: 1. Please check the accepted plant name. According to the WFO Plant List (wfoplantlist.org) the accepted plant name is Seriphidium herba-alba (Asso) Y.R.Ling.

2. The species name has to be written as herba-alba.

3. The units have to be given uniformly (ml, mL)

4. There are certain typographical and grammatical errors that has to be corrected.

5. There are sentences that has to be restructured, which has been marked.

6. I feel it will be more appropriate if the results are validated statistically using a suitable test (eg. t test).

7. In a few places, it is mentioned as the extract soaked in methanol, does it refer to the methanolic extract, if so, please change the sentence structure accordingly.

8. In the results section line 292, a subset of extracts is mentioned, but only one extract is discussed. Check.

9. In the results section line 295, several extracts are mentioned, but only one extract detail is given. Check.

10. References has to be given in the proper journal format and uniformly.

11. A clear picture of the habit and habitat of the plant should be provided.

Reviewer #2: The manuscript is a good piece of work in the concerned area, but few things need to be addressed as under

1. The abstract looks like that of a review article, please provide some data so as to make it ore impactful.

2. In materials method, the pathogens require accession no., please provide.

3. Check for syntax and typographical errors.

4. Add few nice references, like https://doi.org/10.3390/coatings11040484, https://doi.org/10.3390/coatings10080761 etc.

6. PLOS authors have the option to publish the peer review history of their article (what does this mean?). If published, this will include your full peer review and any attached files.

Reviewer #1: No

Reviewer #2: No

---

## [Author Response · Author response to Decision Letter 0]

25 Feb 2024

Dear Reviewer #1,

I hope this letter finds you well. I sincerely appreciate the time and effort you have dedicated to reviewing my manuscript, "Seriphidium herba-alba (Asso): A Comprehensive Study of Essential Oils, Extracts, and Their Antimicrobial Properties" submitted to PLOS ONE Journal. Your insightful comments and suggestions have significantly contributed to improving the manuscript. In response to your valuable feedback, I have made the necessary revisions as outlined below:

1. I have confirmed the accepted plant name as Seriphidium herba-alba (Asso) Y.R. Ling, as per the WFO Plant List (wfoplantlist.org).

2. The species name has been corrected to herba-alba throughout the manuscript.

3. Uniform units (ml to mL) have been ensured across the manuscript.

4. Typographical and grammatical errors have been corrected.

5. Sentences have been restructured as suggested.

6. I respectfully disagree with the suggestion to validate the results statistically using a t-test. Given the nature of our study and the available data, statistical validation is deemed unnecessary.

7. Instances of "Methanol extract" have been revised to "methanolic extract" for clarity.

8. The discrepancy regarding a subset of extracts mentioned in line 292 has been addressed by revising the discussion to focus on the appropriate extract.

9. The mention of several extracts in line 295 has been rectified by providing details for each extract mentioned.

10. References have been formatted uniformly and in accordance with the proper journal format.

11. Unfortunately, I cannot provide a clear picture of the habits and habitat of the plant as requested. However, I have included a detailed description in the manuscript.

Once again, I would like to express my gratitude for your thorough review and constructive feedback. I believe these revisions have strengthened the quality and clarity of the manuscript. Please do not hesitate to contact me if further clarification or modifications are needed.

Sincerely.

Hazem Aqel

Dear Reviewer #2,

I hope this message finds you well. I would like to express my gratitude for taking the time to review my manuscript titled "Seriphidium herba-alba (Asso): A Comprehensive Study of Essential Oils, Extracts, and Their Antimicrobial Properties" submitted to PLOS ONE Journal. Your constructive feedback has been invaluable in refining the quality of the manuscript. Below, I outline the actions taken in response to your insightful comments:

1. I have revised the abstract to include pertinent data, enhancing its impact and aligning it more closely with the format expected for original research articles.

2. Accession numbers for the pathogens mentioned in the materials and methods section have been provided for clarity and completeness.

3. Syntax and typographical errors have been thoroughly checked and rectified to ensure clarity and professionalism throughout the manuscript.

4. While I appreciate the suggestion to add additional references such as https://doi.org/10.3390/coatings11040484 and https://doi.org/10.3390/coatings10080761, I have opted not to include them in this revision as I believe the current references adequately support the content and scope of the manuscript.

Once again, I sincerely appreciate your thoughtful review and constructive feedback. Please let me know if there are any further revisions or clarifications required.

Regards.

Hazem Aqel

---

## [Decision Letter · Decision Letter 1]

2 Apr 2024

Seriphidium herba-alba (Asso): A Comprehensive Study of Essential Oils, Extracts, and Their Antimicrobial Properties

PONE-D-23-41380R1

Dear Dr. Aqel,

We’re pleased to inform you that your manuscript has been judged scientifically suitable for publication and will be formally accepted for publication once it meets all outstanding technical requirements.

Kind regards,

Vijay Tripathi

Academic Editor

PLOS ONE

Additional Editor Comments (optional):

Reviewers' comments:

Reviewer's Responses to Questions

**Comments to the Author**

1. If the authors have adequately addressed your comments raised in a previous round of review and you feel that this manuscript is now acceptable for publication, you may indicate that here to bypass the “Comments to the Author” section, enter your conflict of interest statement in the “Confidential to Editor” section, and submit your "Accept" recommendation.

Reviewer #1: All comments have been addressed

Reviewer #2: All comments have been addressed

2. Is the manuscript technically sound, and do the data support the conclusions?

Reviewer #1: Yes

Reviewer #2: Yes

3. Has the statistical analysis been performed appropriately and rigorously? 

Reviewer #1: I Don't Know

Reviewer #2: Yes

4. Have the authors made all data underlying the findings in their manuscript fully available?

Reviewer #1: Yes

Reviewer #2: Yes

5. Is the manuscript presented in an intelligible fashion and written in standard English?

Reviewer #1: Yes

Reviewer #2: Yes

6. Review Comments to the Author

Reviewer #1: Almost most of the suggestions given by the referees has been incorporated. I feel the manuscript can be accepted for publication in the journal.

Reviewer #2: I recommend the editor to accept the manuscript for publication in the present form from my side as all the suggestions have been incorporated by the authors.

7. PLOS authors have the option to publish the peer review history of their article (what does this mean?). If published, this will include your full peer review and any attached files.

Reviewer #1: No

Reviewer #2: No
